# Metabolic Associated Fatty Liver Disease as a Risk Factor for the Development of Central Nervous System Disorders

**Sayuri Yoshikawa, Kurumi Taniguchi, Haruka Sawamura, Yuka Ikeda, Tomoko Asai** [ID]**, Ai Tsuji and Satoru Matsuda** *[ID]

Department of Food Science and Nutrition, Nara Women's University, Kita-Uoya Nishimachi, Nara 630-8506, Japan
* Correspondence: smatsuda@cc.nara-wu.ac.jp; Tel./Fax: +81-742-20-3451

**Abstract:** MAFLD/NAFLD is the most ordinary liver disease categorized by hepatic steatosis with the increase of surplus fat in the liver and metabolic liver dysfunction, which is associated with bigger mortality and a high medical burden. An association between MAFLD/NAFLD and central nervous system disorders including psychological disorders has been demonstrated. Additionally, MAFLD/NAFLD has been correlated with various types of neurodegenerative disorders such as amyotrophic lateral sclerosis or Parkinson's disease. Contrasted to healthy controls, patients with MAFLD/NAFLD have a greater prevalence risk of extrahepatic complications within multiple organs. Dietary interventions have emerged as effective strategies for MAFLD/NAFLD. The PI3K/AKT/mTOR signaling pathway involved in the regulation of Th17/Treg balance might promote the pathogenesis of several diseases including MAFLD/NAFLD. As extrahepatic complications may happen across various organs including CNS, cooperative care with individual experts is also necessary for managing patients with MAFLD/NAFLD.

**Keywords:** NAFLD; MAFLD; Th17 cells; Treg cells; PI3K; AKT; /mTOR; signaling pathway; gut microbiota; probiotics

## 1. Introduction

Non-alcoholic fatty liver disease (NAFLD) is the most ordinary liver disease with a worldwide incidence of approximately 25% [1]. NAFLD is now related to a heavy socioeconomic burden. The feature of the disease is hepatic steatosis with the accumulation of surplus fat in the liver and metabolic liver dysfunction. Therefore, it has been suggested that NAFLD should be retitled as metabolic disorder-associated fatty liver disease (MAFLD) [2,3]. Here, we use the term MAFLD instead of NAFLD. MAFLD is often supposed to be practically asymptomatic. However, many MAFLD patients complain of exhaustion, which may disturb their quality of life (QOL). Impaired QOL in patients with MAFLD may be associated with depression and fatigue, and together they might hinder various physical activities. Published information could also support the role of inflammation in both depression and MAFLD, suggesting that both illnesses are correlated [4]. In addition, abnormalities in fat accumulation have previously been identified in patients with motor neuron diseases such as amyotrophic lateral sclerosis (ALS) [5]. Fatty liver disease may characterize a non-neuronal clinical condition of various forms of motor neuron disease [6]. Similarly, it has been reported an increased incidence of MAFLD in patients with spinal and bulbar muscular atrophy, in which distinct changes of hepatic gene expression in the patient of spinal and bulbar muscular atrophy have been shown compared to others [7]. Most studies on the association of MAFLD with these neurological disorders in the central nervous system (CNS) or brain disorders have been aimed at the investigation of pathophysiological bases of the comorbidity. Previous studies have reported associations between gamma-glutamyltransferase levels and the onset of

neurodegenerative diseases such as dementia [8]. Cross-sectional studies have emphasized a strong positive association between sedentary time and MAFLD [9], proposing that patients with motor neuron disease may promote the development of MAFLD owing to reduced general mobility. However, the MAFLD phenotype might precede findings of motor dysfunction [10]. Additional research is needed to understand the occurrence of MAFLD in neurological disorders and to characterize the relationship of this process with the underlying disease mechanisms.

It has been described that obesity is associated with liver disorders such as fatty liver [11]. Obesity is the most prevalent risk factor for MAFLD [12]. Obesity is a medical condition in which excess body fat increases to the point where it damagingly affects the health of the host. Poor vitality in MAFLD patients might be along with the presence of metabolic comorbidity such as obesity and then significant fibrosis might predict more depressive symptoms [13]. In general, obesity is also related to inflammation. Obesity might be characterized by chronic inflammation with undyingly increased oxidative stresses through the production of various adipokines from adipose tissue in obesity. Much published literature has established that obesity could increase proinflammatory cytokine expression and/or decreases the production of anti-inflammatory cytokine [14]. It is well recognized that obesity is involved in inflammatory and autoimmune diseases associated with interleukin-17 (IL17) producing Th17 cells, which is also potential pathogenesis of several diseases including type 2 diabetes [15]. Fructose is broadly used in processed foods and numerous beverages. Excessive intake of fructose is well known to be related to diabetes, which could also trigger hepatic steatosis and dyslipidemia, leading to the development of metabolic syndrome including obesity and MAFLD [16].

## 2. Th17/Treg Balance Involved in MAFLD and/or Psychiatric Disorders

MAFLD as well as particularly its more serious form with non-alcoholic steatohepatitis (NASH) could develop from metabolic syndrome, type 2 diabetes, and obesity, which may develop a prominent cause of liver fibrosis [17]. Interestingly, an elevation in the number of Th17 cells has been repetitively detected in the livers of MAFLD mouse models [18]. Likewise, raised number of Th17 cells in circulation and/or in the liver are also detected in MAFLD/NASH patients [19]. It is familiar that CD4 positive T cells have diverse subset cells such as Th1, Th17, and regulatory T (Treg) cells, which are categorized by expression of corresponding diverse cytokines [20]. Th17 cells could stimulate liver inflammation and/or liver fibrosis plausibly by playacting on liver cells mainly the Stellate cells and/or Kupffer cells to speed up the liver fibrotic process [21]. Treg cells might play crucial roles in controlling immune homeostasis. A decrease in cell numbers in hepatic Treg cells has been also observed in animal models of MAFLD [18,22]. Furthermore, the numbers of circulating Treg cells and/or resting Treg cells in the liver may be lower in MAFLD patients than that in healthy controls with an even more vigorous decrease in patients with NASH [19,23]. Treg cells might have double roles in NASH because of their spatial and/or time-based actions in the development of this disease. Therefore, the Th17/Treg ratio in the liver might be valuable in classifying patients with MAFLD/NASH from those with light-degreed or simple steatosis. Th17/Treg balance could also affect the levels of various inflammatory cytokines in MAFLD patients [24] (Figure 1).

The Th17/Treg balance has been suggested to play an important role in the pathophysiology of depression. In fact, major depressive disorder has been revealed to bring a substantial increase in the cell number of peripheral Th17 cells and an apparent decrease in Treg cell numbers, exhibiting an imbalance of the Th17/Treg ratio compared to that of healthy controls [25]. Amazingly, the infusion of Th17 cells could provoke a depression-like behavior in a mouse model with chronic restraint stresses [26]. Possibly, the development of depressive symptoms also results from altered Th17 cell numbers. Similarly, major depressive disorder patients may show an expansion of circulating Treg cells [27]. There is growing interest in the specific role of Th17 cells and/or Treg cells in the pathogenesis of CNS disorders and/or neurodegenerative diseases [28]. Th17 cells could guide the

irregular inflammatory response including the excessive activation of microglia and/or the recruitment of other immune cells to CNS for the progression of the disease [29]. For example, peripheral Th17 cell-mediated inflammatory immune responses in the immune structure of ALS patients might be confidently interrelated with the disease level and/or progression [30]. However, the precise mechanisms of Th17 cells and/or Treg cells as well as their linked cytokines in the neuropathology of those neurological disorders have not been elucidated totally.

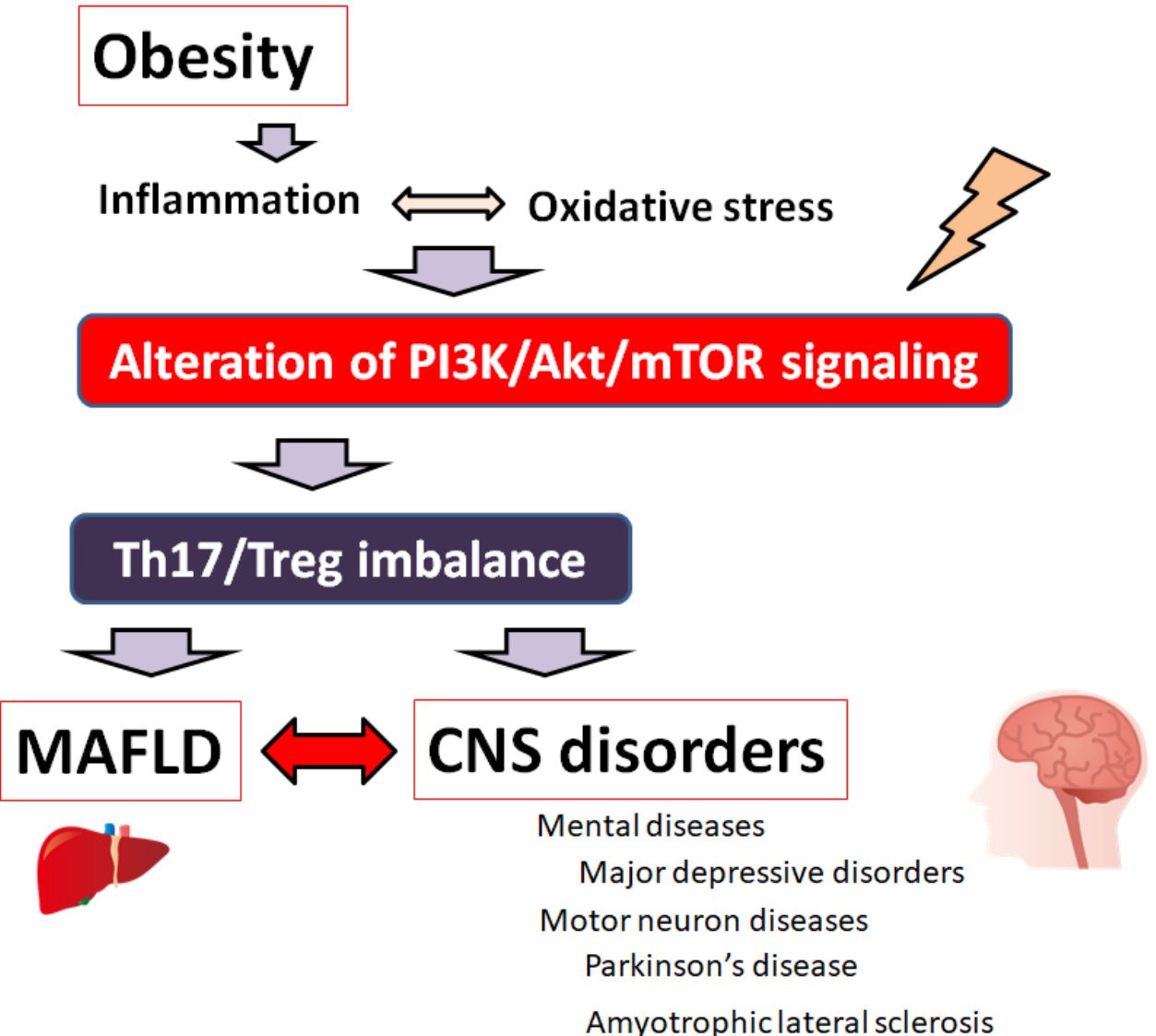

**Figure 1.** A hypothetical schematic representation and overview of the pathogenesis of MAFLD and various brain disorders including mental diseases and/or motor neuron diseases. Obesity, inflammation, and/or oxidative stress could lead to the alteration of PI3K/AKT/mTOR signaling following contribution to the imbalance of Th17/Treg ratio of immune cells, which might eventually lead to the pathogenesis both of MAFLD and brain disorders. Note that several significant things have been omitted for clarity.

### 3. PI3K/AKT/mTOR Signaling Pathway Involved in the Regulation of Th17/Treg Balance of Various Diseases

Hepatic infiltration of Th17 cells might be critical for the NASH triggering and/or development of liver fibrosis [31]. Therefore, the maturation of Th17 cells and the Th17/Treg balance axis are major contributing factors to the pathogenesis of MAFLD as well as NASH.

Regulation of Th17 cells or Treg cells may be regulated through the PI3K/AKT/mTOR intracellular signaling pathway [32] (Figure 1). For example, programmed death-ligand 1 (PD-L1) is involved in regulating Th17/Treg cell balance in ulcerative colitis (UC) by blocking the activation of the PI3K/AKT/mTOR signaling pathway [33]. In addition, Th17 differentiation may be closely related to the inflammatory response of synoviocytes through PI3K/AKT/mTOR signaling pathway [34]. One of the microRNAs, miR-151-5p, could balance Th17/Treg by modulating the PI3K/AKT/mTOR signaling pathway [35]. It has also been shown that upregulation of miR-151-5p could alter the Th17/Treg ratio via the activation of PI3K/AKT/mTOR signaling [36]. The PI3K/AKT/mTOR signaling pathway is involved in fundamental cellular processes including apoptosis, metabolism, cycle, autophagy, and survival to play a significant role in the homeostasis of various cells and/or organs [37]. For example, the PI3K/AKT/mTOR pathway could be involved even in the retrieval of ovarian function by changing the ratio of Th17/Tc17 and Th17/Treg cells [38]. Rapamycin-mediated blockage of mTOR activation may restrain T-cell proliferation for decreased Th17/Treg ratios [39]. Similarly, mTOR activation may be positively correlated with the loss of Th17/Treg balance [40]. Th17/Treg balance could be modulated by regulating the PI3K/AKT/mTOR signaling pathway in immune cells [41]. Consistently, the Treg/Th17 imbalance might be associated with the activation of PI3K/AKT/mTOR signaling in peripheral blood mononuclear cells [42]. In addition, the PI3K/AKT/mTOR signaling could restore the Th17/Treg balance in chronic obstructive pulmonary disease [43]. Sarcoidosis is a systemic granulomatous disease associated with the Treg cells paradigm, in which PI3K/AKT/mTOR signaling is critical for the optimal Treg responses [44]. The imbalance of Th17/Treg is a critical factor even in asthma pathogenesis, which could be improved by the inhibition of the PI3K/AKT/mTOR pathway for airway protection [45]. The dynamic equilibrium between Th17/Treg immune cells via the PI3K/AKT/mTOR pathway has been shown at the maternal-fetal interface [46]. Modulation of Th17/Treg balance via the PI3K/AKT/mTOR signaling could prevent cartilage and/or bone destruction [47]. An imbalance of Th17/Treg has been found in patients with intracranial aneurysms, which could be inhibited through suppression of the PI3K/AKT/mTOR and NFκB signaling [48]. High-fat and high-fructose diet could induce metabolic syndrome and NASH, which might coincide with an increase in hepatic Th17 cells [49]. A high-fat diet may also exacerbate depressive-like behavior [50]. In these ways, the PI3K/AKT/mTOR pathway involved in the regulation of Th17/Treg balance might influence the pathogenesis of various diseases including MAFLD. (Figure 2).

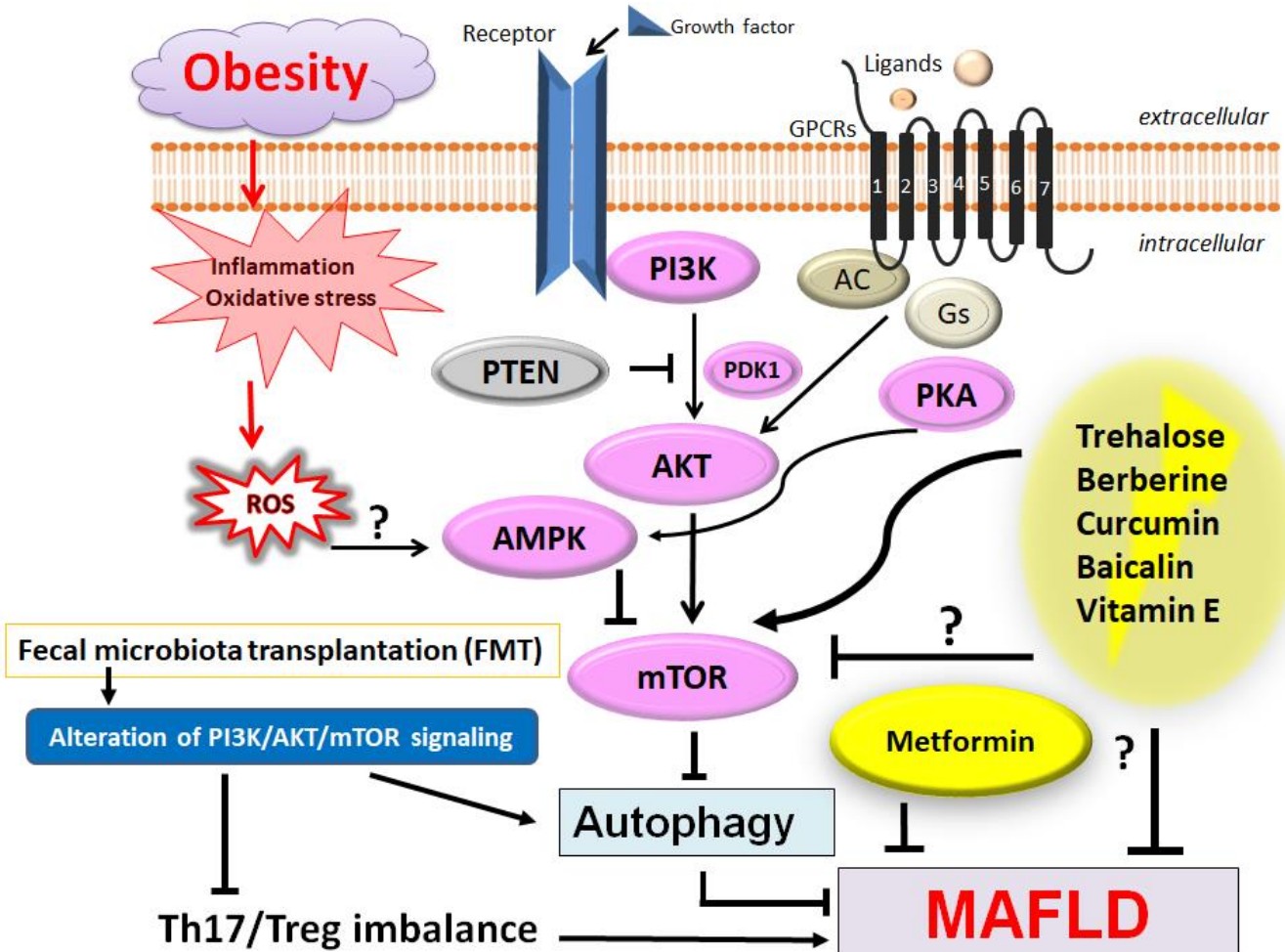

**Figure 2.** Several modulator molecules linked to the PI3K/AKT/mTOR signaling pathway have been shown. Example compounds from natural sources known to act on the AMPK/PI3K/AKT/mTOR and/or autophagy-related signaling pathway are also shown. Reactive oxygen species (ROS), inflammation, and/or autophagy might be all involved in the pathogenesis of MAFLD. Fecal microbiota transplantation (FMT) could be effective against the progression of MAFLD. Arrowhead indicates stimulation whereas the hammerhead shows inhibition. Note that several important activities such as cytokine induction or anti-inflammatory reaction have been omitted for clarity. Abbreviation: AMPK, adenosine monophosphate-activated protein kinase; mTOR, mammalian/mechanistic target of rapamycin; PI3K, phosphoinositide-3 kinase; PKA, protein kinase A; PTEN, phosphatase and tensin homolog deleted on chromosome 10.

## 4. Therapeutic Strategies for Patients with MAFLD and/or Various CNS Disorders

Some satisfying strategies have been recognized for patients with MAFLD. Emerging evidence recommends that anti-diabetic medications could decrease fatty accumulation and/or decline liver enzyme levels in MAFLD [51]. For example, metformin is a biguanide anti-diabetic drug that has been utilized to treat patients with type 2 diabetes. Metformin has been proven to have an outstanding therapeutic effect on MAFLD [52]. Metformin could also inhibit the inflammatory reaction that can regulate Th17 cells and Treg cells in a rheumatoid arthritis mouse model [53]. In addition, metformin has been shown to have an anti-inflammatory property in a mouse model of inflammation-associated tumors [54]. Additionally, metformin could ameliorate arthritic symptoms by suppressing Th17 differentiation [55].

Dietary interventions have emerged as effective palliative strategies for MAFLD. For example, studies have shown the benefit of antioxidants such as vitamin E in various

common foods on fatty liver progression [56]. For another example, berberine could inhibit the proliferation of Th17 cells and could also promote the differentiation of Treg cells via the PI3K/AKT/mTOR signaling [57]. Hence, berberine has been widely used to treat MAFLD [58]. Oxyberberine, a gut microbiota-mediated oxidative metabolite of berberine, has been also identified as effective on MAFLD [59]. In addition, curcumin has effectively alleviated colitis in mice with type 2 diabetes mellitus by restoring the homeostasis of the Th17/Treg ratio and improving the composition of the intestinal microbiota [60]. Supplementation of curcumin has an advantageous effect on liver findings, reduced serum liver enzymes, total cholesterol, and body mass index (BMI) in participants with MAFLD [61]. Tetrahydrocurcumin could attenuate hepatic lipogenesis in an adenosine monophosphate-activated protein kinase (AMPK)-dependent manner suggesting a potential treatment for MAFLD [62]. Similarly, dihydrocurcumin could improve hepatocellular glucose uptake by increasing the protein expression levels of PI3K/AKT [63]. Baicalin, an extract from *Scutellaria baicalensis Georgi*, may play a beneficial role by mediating downstream immune response pathways brought by oxidative stresses and/or inflammation, in which PI3K/AKT/mTOR signaling might be a key factor associated with the remedial effects of baicalin on MAFLD/NASH [64]. Disaccharide trehalose might provide structure-specific effects on cellular energy production and hepatic fat accumulation, suggesting a health potential for the treatment of MAFLD [65] (Figure 2). Amazingly, trehalose has been revealed as an attractive candidate to prevent and modify the progression of Parkinson's disease [66].

Microbiota in a body may participate in the pathogenesis of MAFLD by regulating metabolic pathways [67]. The progression of MAFLD is also closely related to some microbiota in the body. For example, *porphyromonas gingivalis*, the main pathogen for periodontal disease, could participate in the development of MAFLD via the Th17/Treg imbalance induced by disordered microbial metabolisms [68]. High-fat diet-related microbiota dysbiosis might be responsible for a decreased number of Th17 cells [69]. An increased abundance of Treg-inducing bacteria that could also stimulate the Treg activity in the colon, might in turn down-regulate the inflammatory signals in the liver [70]. In general, targeting Treg cells could act as a favorable prognostic pointer by modulating steatosis during the pathogenesis of MAFLD and MAFLD-associated hepatocellular carcinoma [71]. Interestingly, trehalose is vilified for its putative microbial effects, which are potent therapeutic actions of trehalose without adversely affecting host microbial communities [72].

Some beneficial methods based on the renovation of gut microbiota conformation, including probiotics and/or fecal microbiota transplantation (FMT), as well as targeted gut microbiota-associated signaling pathways, might present novel visions into the treatment for MAFLD patients [73]. Interestingly, FMT could also show anti-depressant activity [74]. Research on the microbiota-gut-brain axis in major depressive disorder is promising to develop and/or progress novel treatment, which is currently accepted as an indispensable part of the adjustment and/or the maintenance of homeostasis in systemic metabolism [75]. FMT might be also a possible intervention to alter the immunological response to ALS and/or the disease development [76].

## 5. Future Perspectives

MAFLD has developed a main public health concern as its progression increases the risks of multisystem morbidity and mortality. MAFLD is characterized by diffuse hepatic alveolar steatosis and fat stored in liver lobules, with the exception of alcohol and/or other certain liver-damaging factors including NASH [77]. It is well recognized that MAFLD is a systemic disorder with variations in genetic background, metabolic characteristics, dietary habits, lifestyles as well as environmental risks, which could altogether contribute to the pathogenesis of MAFLD [78]. Therefore, extrahepatic complications of MAFLD may include various psychological dysfunction, obstructive sleep apnea syndrome, extrahepatic malignancies such as colorectal cancer, and/or polycystic ovarian syndrome [79]. In particular, major depressive disorder may be highly associated with MAFLD, which might

have complex pathogenic mechanisms [80]. The prevalence of MAFLD is about 50% among people with depression [81]. Patients with MAFLD have a prevalence of about 18 % in mental disorders [82]. It has been described that the brain volumes of white and gray matter have been decreased in patients with MAFLD compared with those of control subjects, which might be associated with a greater risk of depression in patients with MAFLD [83]. Long-term psychological stress might play an imperative role in introducing and/or arbitrating the occurrence of diseases [84]. Unfortunately, MAFLD and major depressive disorder mediate and promote the progression of each other [85]. Consequently, an increase in the occurrence of major depressive disorder is also an imperative public health concern. MAFLD patients with major depressive disorder have a reduced response to the typical care for MAFLD, involving predominantly lifestyle alterations [86]. Interestingly, trehalose may have antidepressant-like properties [87] as well as anti-hepatic-steatosis properties [88]. Hepatic steatosis is also a frequent finding in ALS [89]. Inflammations and/or oxidative stresses have been supposed to be key factors for the pathogenesis of these diseases [90]. Additionally, trehalose might be a valuable add-on therapy in combination with other ALS treatment options to improve symptoms in early-stage of ALS [91]. On the contrary, risperidone, a second-generation antipsychotic drug used for the treatment of schizophrenia and/or major depressive disorder, may exacerbate MAFLD in obese mice [92].

A number of studies have shown that an imbalance of Th17/Treg cells may significantly contribute to the occurrence and/or progression of various inflammatory diseases such as inflammatory bowel disease [93]. The imbalance might be also involved in neurodegenerative pathology including multiple sclerosis [94]. Therefore, the balance between Th17 and Treg cells as well as the related cytokines is crucial, which could be achieved by the regulation of the PI3K/AKT/mTOR signaling pathway [95]. In addition, it has been suggested a critical association between the PI3K/AKT/PTEN signaling pathway and MAFLD [96]. As shown here, MAFLD connected with obesity might be also associated with depression and anxiety-related behavior. This may be accompanied by astrocytic and/or microglial metabolic alterations including higher oxygen consumption, suggesting that the early stages of MAFLD might be related to diet-induced encephalopathy [97]. Not only liver specialists, but also patients with MAFLD, should be conscious of increased risks for various diseases including CNS disorders. Since many complications may potentially occur across various organs including CNS, cooperative care with individual experts is also essential for the good management of patients with MAFLD.

## 6. Conclusions

The Th17/Treg balance via the regulation of PI3K/AKT/mTOR signaling pathway has been suggested to play important roles in the pathophysiology of MAFLD. Since MAFLD is a systemic disorder with differences in genetic background and/or metabolic characteristics, MAFLD may have various complications with CNS disorders including several psychological dysfunctions. Therefore, cooperative care with medical and scientific experts might be indispensable for the good management of patients with MAFLD.

**Author Contributions:** Conceptualization, S.Y. and S.M.; original draft preparation and editing, S.Y., K.T., H.S., Y.I., T.A., A.T. and S.M.; visualization, S.Y. and S.M.; supervision, S.M. Each author (S.Y., K.T., H.S., Y.I., T.A., A.T. and S.M.) has participated sufficiently in this work of drafting the article and/or revising the article for the important rational content. Then, all authors gave final approval of the version to be submitted. All authors have read and agreed to the published version of the manuscript.

**Funding:** This research received no external funding.

**Institutional Review Board Statement:** Not applicable.

**Informed Consent Statement:** Not applicable.

**Data Availability Statement:** Not applicable.

**Conflicts of Interest:** The authors declare no conflict of interest.

**Abbreviations**

| | |
|---|---|
| ALS | amyotrophic lateral sclerosis |
| AMP | Adenosine monophosphate |
| AMPK | AMP-activated protein kinase |
| BMI | body mass index |
| CNS | central nervous system |
| FMT | fecal microbiota transplantation |
| NAFLD | non-alcoholic fatty liver disease |
| MAFLD | metabolic-associated fatty liver disease |
| NASH | non-alcoholic steatohepatitis |
| mTOR | mammalian/mechanistic target of rapamycin |
| PI3K | phosphoinositide-3 kinase |
| PKA | protein kinase A |
| PTEN | phosphatase and tensin homologue deleted on chromosome 10 |
| QOL | quality of life |
| ROS | reactive oxygen species |
| UC | ulcerative colitis |

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
