# Peer review of "Metabolic Associated Fatty Liver Disease as a Risk Factor for the Development of Central Nervous System Disorders"

_livers, doi:10.3390/livers3010002_

Round 1

Reviewer 1 Report

Title: Metabolic associated fatty liver disease and risk of central nervous system disorders

The author has reviewed the association or comorbidities of neuronal disorders in metabolic-associated fatty liver diseases.  

Comments:

  1. The manuscript is too verbose.
  2. The author is more focused on the upregulation of Th17 cells and the alteration of Th17/TReg cells in metabolic-associated liver diseases, which does not directly support this work's title. In addition, this manuscript has no clear-cut explanation with the literature supporting to elucidate the actual mechanism of Th17 cells in neuronal damage.
  3. The author has collectively described the mental disorders in metabolic-associated liver diseases with little literature; however, there should be a more elaborated explanation with citing articles needed to make it more attractive.

Author Response

  1. The manuscript is too verbose.

We have improved the text for the better readability to readers.

  1. The author is more focused on the upregulation of Th17 cells and the alteration of Th17/TReg cells in metabolic-associated liver diseases, which does not directly support this work's title. In addition, this manuscript has no clear-cut explanation with the literature supporting to elucidate the actual mechanism of Th17 cells in neuronal damage.

According to the suggestion, we have altered the title of this manuscript. In addition, we have added some explanation and literature for the mechanism in the text of section 5.

  1. The author has collectively described the mental disorders in metabolic-associated liver diseases with little literature; however, there should be a more elaborated explanation with citing articles needed to make it more attractive.

Thank you so much. We have tried to make this manuscript more attractive with the improvement of figures and text citing additional literatures.

Reviewer 2 Report

Authors clearly present how MAFLD Links CNS disorder by altering the Th17/Treg balance through PI3K/AKT/mTOR signaling pathways. Overall, the review is well described and well structured.

In my opinion authors can include a section about how obesity induce inflammation and this inflammation is the trigger for all downstream process.

Otherwise, I fell review is really interesting.

Author Response

Authors clearly present how MAFLD Links CNS disorder by altering the Th17/Treg balance through PI3K/AKT/mTOR signaling pathways. Overall, the review is well described and well structured.

In my opinion authors can include a section about how obesity induce inflammation and this inflammation is the trigger for all downstream process.

Otherwise, I fell review is really interesting.

Thank you for the good evaluation on our manuscript. We have added some explanation about the inflammatory signaling from obesity in the text of section 1 and 5.

Reviewer 3 Report

In this manuscript, the authors reviewed selected key discoveries related to the possible connections between MAFLD/NAFLD and central nervous system disorders, highlighting the involvement of the PI3K/AKT/mTOR signaling pathway and Th17/Treg imbalance in the pathogenesis of MAFLD/NAFLD -associated brain disorders. This opinion thus provides useful insights for both the scientific community and the industry.

However, the author should consider addressing the following issues in this manuscript:

Major points:

1. The authors should consider reviewing and discussing some literature on hepatic encephalopathy which directly connects various liver failures with central nervous systems.

2. Fig.2 is too messy, it is not clear whether this is a summarization or a hypothesis. The relationships between several components in the figure are not clear. e.g. autophagy and MAFLD. The authors might consider reorganizing the figure completely and adding a figure title.

3. The authors should consider elaborating on the Future Perspective section, proposing (1). the avenues for the pathological study involving the PI3K/AKT/mTOR signaling pathway and  Th17/Treg balance, (2) possible technical challenges for future research in this field, (3) how modulation of the PI3K/AKT/mTOR signaling pathway and Th17/Treg imbalance could benefit the development of future therapies against MAFLD-related central nervous system disorders.  

Minor points:

1. For Fig 1, the authors should consider checking and re-organizing the positions of obesity, inflammation, and oxidative stress.  Is oxidative stress derived from obesity and inflammation?

2. The authors should not stating other opinions other than those related to the scientific topic. e.g. “We are sure that Japanese sweets containing a little bit trehalose are of particular good taste.” is not very appropriate.

3. The authors should consider changing the title of the paper to accurately reflect its content.

To summarize, I agree that this paper may be relevant for publication in Livers. However, this manuscript could be improved by addressing the above issues, I, therefore recommend reconsideration of the manuscript after major revision.

Author Response

In this manuscript, the authors reviewed selected key discoveries related to the possible connections between MAFLD/NAFLD and central nervous system disorders, highlighting the involvement of the PI3K/AKT/mTOR signaling pathway and Th17/Treg imbalance in the pathogenesis of MAFLD/NAFLD -associated brain disorders. This opinion thus provides useful insights for both the scientific community and the industry.

However, the author should consider addressing the following issues in this manuscript:

Major points:

  1. The authors should consider reviewing and discussing some literature on hepatic encephalopathy which directly connects various liver failures with central nervous systems.

 As for the hepatic encephalopathy, we have added some explanation in the text of section 5.

  1. Fig.2 is too messy, it is not clear whether this is a summarization or a hypothesis. The relationships between several components in the figure are not clear. e.g. autophagy and MAFLD. The authors might consider reorganizing the figure completely and adding a figure title.

According to this suggestion, Fig. 2 has been improved.

  1. The authors should consider elaborating on the Future Perspective section, proposing (1). the avenues for the pathological study involving the PI3K/AKT/mTOR signaling pathway and  Th17/Treg balance, (2) possible technical challenges for future research in this field, (3) how modulation of the PI3K/AKT/mTOR signaling pathway and Th17/Treg imbalance could benefit the development of future therapies against MAFLD-related central nervous system disorders.  

 Yes, we have added the context relating to above 1, 2, 3, in the text of Section 5.

Minor points:

  1. For Fig 1, the authors should consider checking and re-organizing the positions of obesity, inflammation, and oxidative stress.  Is oxidative stress derived from obesity and inflammation?

Yes, absolutely. The concept has been shown in several literatures such as below.

Obesity is characterized by chronic low grade inflammation with permanently increased oxidative stress [doi: 10.3390/ijms16010378].

Adipokines from adipose tissue in obesity could induce the production of reactive oxygen species (ROS), generating a process known as oxidative stress [doi: 10.3390/ijms12053117]. 

Obesity can induce systemic oxidative stress through various biochemical mechanisms, such as superoxide generation from NADPH oxidases, oxidative phosphorylation, and glyceraldehyde auto-oxidation [ doi: 10.1089/met.2015.0095].

  1. The authors should not stating other opinions other than those related to the scientific topic. e.g. “We are sure that Japanese sweets containing a little bit trehalose are of particular good taste.” is not very appropriate.

 We have deleted the sentence, according to this suggestion.

  1. The authors should consider changing the title of the paper to accurately reflect its content.

Yes. We have change the title, according to this suggestion.

To summarize, I agree that this paper may be relevant for publication in Livers. However, this manuscript could be improved by addressing the above issues, I, therefore recommend reconsideration of the manuscript after major revision.

Thank you so much for good advices to improve the manuscript.

Round 2

Reviewer 1 Report

Title: Metabolic-associated fatty liver disease as a risk factor for the development of central nervous system disorders

In this current manuscript, the author highlighted metabolic-associated liver diseases as a risk factor for developing central nervous system disorder through the imbalance in T17 and TReg cells. The author has well revised the manuscript and incorporated the suggestion. 

Author Response

Title: Metabolic-associated fatty liver disease as a risk factor for the development of central nervous system disorders

In this current manuscript, the author highlighted metabolic-associated liver diseases as a risk factor for developing central nervous system disorder through the imbalance in T17 and TReg cells. The author has well revised the manuscript and incorporated the suggestion. 

Thank you so much for the good evaluation on the revised version of our manuscript.

Reviewer 3 Report

In this version, the authors have made relevant modifications or provided reasonable explanations regarding the majority of issues mentioned in the previous comments. Thus, the manuscript is being improved considerably. I, therefore recommend acceptance of the manuscript after proofreading and minor revision.

Author Response

In this version, the authors have made relevant modifications or provided reasonable explanations regarding the majority of issues mentioned in the previous comments. Thus, the manuscript is being improved considerably. I, therefore recommend acceptance of the manuscript after proofreading and minor revision.

Thank you so much for the good evaluation on the revised version of our manuscript. In addition, we have improved the manuscript for the better readability.